# Advanced and Smart Textiles during and after the COVID-19 Pandemic: Issues, Challenges, and Innovations

**DOI:** 10.3390/healthcare11081115

**Published:** 2023-04-13

**Authors:** Aleksandra Ivanoska-Dacikj, Yesim Oguz-Gouillart, Gaffar Hossain, Müslüm Kaplan, Çağlar Sivri, José Vicente Ros-Lis, Daiva Mikucioniene, Muhammad Usman Munir, Nuray Kizildag, Serkan Unal, Ivo Safarik, Esra Akgül, Nida Yıldırım, Ayşe Çelik Bedeloğlu, Ömer Faruk Ünsal, Gordon Herwig, René M. Rossi, Peter Wick, Pietro Clement, A. Sezai Sarac

**Affiliations:** 1Research Centre for Environment and Materials, Macedonian Academy of Sciences and Arts, Krste Misirkov 2, 1000 Skopje, North Macedonia; 2Department of Building and Urban Environment, Innovative Textile Material, JUNIA, 59000 Lille, France; 3V-Trion GmbH Textile Research, Millennium Park 15, 6890 Lustenau, Austria; 4Department of Textile Engineering, Faculty of Engineering, Architecture and Design, Bartin University, Bartin 74110, Turkey; 5Management Engineering Department, Faculty of Engineering and Natural Sciences, Bahcesehir University, İstanbul 34349, Turkey; 6Centro de Reconocimiento Molecular y Desarrollo Tecnologico (IDM), Unidad Mixta Universitat Politecnica de Valencia, Universitat de Valencia, Departamento de Química Inorgánica, Universitat de València, Doctor Moliner 56, 46100 Valencia, Spain; 7Faculty of Mechanical Engineering and Design, Kaunas University of Technology, Studentu Str. 56, 50404 Kaunas, Lithuania; 8Institute of Nanotechnology, Gebze Technical University, Gebze, Kocaeli 41400, Turkey; 9Integrated Manufacturing Technologies Research and Application Center, Sabanci University, Pendik, Istanbul 34906, Turkey; 10Faculty of Engineering and Natural Sciences, Material Science and Nanoengineering, Sabanci University, Tuzla, Istanbul 34956, Turkey; 11Department of Nanobiotechnology, Biology Centre, ISBB, CAS, Na Sadkach 7, 370 05 Ceske Budejovice, Czech Republic; 12Regional Centre of Advanced Technologies and Materials, Czech Advanced Technology and Research Institute, Palacky University, Slechtitelu 27, 783 71 Olomouc, Czech Republic; 13Department of Industrial Design Engineering, Faculty of Engineering, Erciyes University, Kayseri 38039, Turkey; 14Trabzon Vocational School, Karadeniz Technical University, Trabzon 61080, Turkey; 15Department of Polymer Materials Engineering, Faculty of Engineering and Natural Sciences, Bursa Technical University, Bursa 16310, Turkey; 16Empa, Swiss Federal Laboratories for Materials Science and Technology, Laboratory for Biomimetic Membranes and Textiles, 9014 St. Gallen, Switzerland; 17Empa, Swiss Federal Laboratories for Materials Science and Technology, Laboratory for Particle-Biology Interactions, 9014 St. Gallen, Switzerland; 18Department of Chemistry, Polymer Science and Technology, Faculty of Sciences and Letters, Istanbul Technical University, Istanbul 34469, Turkey

**Keywords:** smart textiles, medical textiles, COVID-19, textiles for diagnostic and prevention of diseases, telemedicine

## Abstract

The COVID-19 pandemic has hugely affected the textile and apparel industry. Besides the negative impact due to supply chain disruptions, drop in demand, liquidity problems, and overstocking, this pandemic was found to be a window of opportunity since it accelerated the ongoing digitalization trends and the use of functional materials in the textile industry. This review paper covers the development of smart and advanced textiles that emerged as a response to the outbreak of SARS-CoV-2. We extensively cover the advancements in developing smart textiles that enable monitoring and sensing through electrospun nanofibers and nanogenerators. Additionally, we focus on improving medical textiles mainly through enhanced antiviral capabilities, which play a crucial role in pandemic prevention, protection, and control. We summarize the challenges that arise from personal protective equipment (PPE) disposal and finally give an overview of new smart textile-based products that emerged in the markets related to the control and spread reduction of SARS-CoV-2.

## 1. Introduction

The global COVID-19 pandemic spread worldwide quickly, affecting many aspects of daily life, public health, the global economy, and social stability. As a solution in the first phase, many countries took action by using personal protective equipment (PPE), especially masks, which were found to be effective in preventing the spread of the virus [1]. Most countries mandated the wearing of masks in public spaces. These efforts included severe measures, such as banning citizens from leaving their homes in many countries and effectively suspending all social and economic activities.

Just over three years have passed since the coronavirus outbreak, as on March 11, 2020, the World Health Organization (WHO) declared the global spread of SARS-CoV-2 a pandemic. More than 550 million people were infected, and more than 6,300,000 died as a consequence of the infection [2].

The epidemiological situation improved and relaxations of COVID-19 rules were implemented worldwide [3]. Still, the emergence of the pandemic irreversibly changed the world, as it is the first pandemic in the digital age. Yet, even the most developed parts of the world were found to be unprepared with no action strategy in place.

Textiles, whether regular, advanced, or smart, played an essential role in tackling this pandemic, mainly as part of technical and medical textiles (e.g., personal protective equipment (PPE), sensors, and telemedicine), which mirrors the importance of textiles in tackling past pandemics [4].

As a major contributor to the European economy, with a current annual turnover of EUR 162 billion and employing over 1.9 million people [5], the textile industry has been hit hard by the COVID-19 crisis due to supply chain disruptions, a drop in demand, liquidity problems, and overstocking. In Europe, production fell by 16.8% in the period between January and April 2020 in comparison with 2019. Nevertheless, the pandemic also created an urgent new demand for technical textiles in the field of personal protective equipment (PPE) and opened a window of opportunity for the sector to rationalize production, revise supply chains, and push for more digitalization. Personal protective equipment with functional textile materials and devices for combating coronavirus infections and smart e-textile wearables for remote diagnosis [5] and monitoring saw rapid development. These technological innovations played an important part in solving the current crisis and might be used as a reference for managing future severe situations.

This pandemic attracted the widespread attention of both the public and the scientific community. Based on a search of Web of Science, up to now, more than 28,000 review papers and more than 230,000 research papers have been published where the “COVID-19” essential word appears in the title, but most of them are outside the textile research and technology field. 

Here, an effort was made to explore and highlight the development of smart and advanced textiles that emerged in response to the outbreak of SARS-CoV-2. The first section showcases the advances in textiles as part of PPE, in particular, fundamental research on mask properties, antiviral functionalization, and characterization, as well as disposal concerns. Subsequently, textile-based developments intended for monitoring and sensing as part of telemedicine are highlighted, with a special focus on electrospun membrane sensors and nanogenerators. Finally, advanced products that managed to surpass the initial development stage and emerge into the market are presented. Combined, these methodologies to assess the challenges and opportunities of the pandemic, as well as the subsequent successful implementation of adapted technology and product establishment, might serve as an example strategy for other industries in similar situations.

## 2. Medical Textiles in Pandemic Control and Prevention

From the cradle to the grave, medical textiles are an indispensable part of our life. Surgical gowns, caps, isolation gowns, masks, and coveralls are key items to achieve success in medical operations, personal protection, or daily hygiene.

Several prevention techniques were devised after it was found that the SARS-CoV-2 coronavirus is transmitted in respiratory particles from an infected person’s mouth or nose [6]. Physical distancing strategies, i.e., maintaining a safe distance between individuals, is the most effective prevention technique against disease spread via the respiratory route [7,8]. When a safe distance is not practicable, personal protective equipment (PPE) is the standard form of self-protection. Masks and respirators are conclusively essential items of PPE. They cover the nose and mouth of the user, and thus, act as a physical barrier to the droplets/particles (carrying virus) during inhalation and exhalation [9,10].

### 2.1. Face Masks—Types and Efficiency

Medical face masks and respirators are of various material types, costs, and protection levels [11]. Medical face masks and respirators come under the category of medical devices; therefore, regulatory authorities around the world already defined their specifications and requirements (see Table 1) and also advised which mask should be used in which specific condition. For instance, type IIR masks are surgical face masks, as they have splash-resistant properties and are tested per the conditions defined in EN 14683: 2019. Type I and type II medical face masks are advised for patients and other persons to reduce the spreading of infections. A type II medical face mask is most promising, as it has a bacterial filtration efficiency of ≥98% with good breathability. 

The material used for the construction and design of medical face masks is advised not to disintegrate, split, or tear during intended use. Medical face masks generally have three layers (spunbond–melt-blown–spunbond) and are made of synthetic fibers, such as polypropylene. Natural fibers are generally not recommended, as they are hydrophilic and may provide sites for bacteria and viruses to grow. Although medical face masks have good filtration, ranging from 95 to 98%, they have a loose fit on the face and air leaks from the sides. To counter this issue, respirators are advised, as they have a close and tight fit to the wearer’s face and decrease the wearer’s exposure to pollutants in the air, such as particles, gases, and vapors. Their specifications and requirements are also defined, as per EN 149:2001, and they may have inhalation or exhalation valves. They come in three different levels of protection: 80% (FFP1), 95% (FFP2, sometimes named N95), and 99% (FFP3) [12] (Table 2).

FFP2 and FFP3 are the most promising respirators, as they filter 95–99% of the particles ranging from 0.1 to 5 μm. Cloth masks are reusable and made of knitted or woven textiles. Despite providing less protection than surgical or non-surgical face masks, they are common in use mainly by the public, and homemade versions are also available [13]. 

Perhaps the essential basic parameters determining face masks’ effectiveness and wear comfort are defined as air permeability and filtration efficiency [14]. Depending on the reference standard, masks usually fall into distinct areas within these categories in accordance with the employed filter layer [15] (Figure 1A). While the general principles of air filtration are known to be complex [16], a high filtration efficiency commonly means a low air permeability and vice versa [17]. Despite the accessibility of advanced materials [18], confusion on actual testing parameters persisted [13,19,20], hence diminishing the confidence of manufacturers in such innovations. Instead, several manufacturers and studies revisited the efficacy of improvised filter materials [19,21]; however, the overall advice remained against material repurposing due to, e.g., biocompatibility and leakage concerns. At the same time, further research showed that non-woven materials commonly provided better air permeability and filtration efficiency combined than woven materials alone [22], as well as strong resistance to mechanical or sweat exposure [23], which explains the wide application of melt-blown non-woven fabrics in air filtration [24]. A quality factor and other models based on structural parameters of woven textiles were developed, which indicated that nanofibers in a less densely packed formation achieve an optimal compromise of both values, as opposed to the macro-fibrous structure of typical woven textiles [25,26] (Figure 1B). Certainly accelerated by the pandemic, electrospinning and similar methods were identified as promising techniques to produce nanofibrous filter materials, which not only satisfy the filtration efficiency and permeability criteria [20,27,28,29] but also offer other benefits, such as transparency [30], reuse after washing or disinfection [31,32], and antimicrobial or antiviral functionalization [33,34,35]. Eventually, during 2021 and convinced by the research and increasing availability [20], a growing number of manufacturers successfully integrated not just melt-blown but also highly efficient nanofibrous non-woven fabrics as filter layers in their community mask compositions (Figure 1C,D) [20,36].

### 2.2. Improvement of Medical Textiles toward Enhanced Anti-Viral Properties

Apart from the general improvement of filtration efficiency and breathability in masks, additional measures targeting SARS-CoV-2 specifically remained imperative. The virus spread was reduced in most countries by enforcing regulations such as social distancing, limiting social activities and events, and compulsory use of protective equipment such as facemasks, which were summarized as non-pharmaceutical measures [37]. However, antiviral functionalization of materials prone to contact with viral particles promised to further reduce the number of infections more reliably [38], with a minimal impact on people’s personal lives and without hampering economic growth. 

#### 2.2.1. Metal-Derived and Carbon-Based Nanomaterials for (Nano)Textile Modification in the Fight against SARS-CoV-2 Infection

It is well known that different types of metal-derived nanoparticles exhibit antiviral properties against various groups of viruses; recently, special attention was paid to checking the effect of various nanomaterials on SARS-CoV-2 [39]. In parallel to this activity, the study and application of metal-derived nanomaterials for (nano)textile modification to develop advanced protective clothing in the fight against SARS-CoV-2 infection are highly important [40]. More detailed information about these topics can be found in recent review papers [41,42,43,44,45]. 

Selected metal-derived nanoparticles, due to their high reactive surface-area-to-volume ratio and unique chemical properties of metal-based nanoparticles, enable their potent inactivation of viruses, thus forming efficient antiseptic coatings to prevent pathogen transmission and infection. Nanoparticles exert their virucidal action through various mechanisms, including inhibition of virus–cell receptor binding, reactive oxygen species oxidation, and destructive displacement bonding with key viral structures [41].

Silver-based nanoparticles (AgNPs) have an extensive history of medical uses due to their unique microbicidal properties. They can be prepared using a wide variety of strategies [46]. AgNPs are known for their high antimicrobial activity, biocompatibility, and low toxicity in eukaryotic cells [41,47]. In one of the in vitro studies that investigated the use of AgNPs against SARS-CoV-2, a large number of AgNPs of different sizes and concentrations were tested. It was observed that particles with a diameter of approximately 10 nm effectively inhibited extracellular SARS-CoV-2 at concentrations ranging between 1 and 10 ppm, while the cytotoxic effect was observed at concentrations of 20 ppm and above. AgNPs potently inhibited the viral entry step by disrupting the viral integrity [48]. In addition, mouthwash and nose rinse containing AgNPs were tested on hospital health workers to prevent SARS-CoV-2 infection. The incidence of SARS-CoV-2 infection was significantly lower in the group testing AgNPs (1.8%) in comparison with the group using regular mouth and nasal rinse (28.2%) [49]. The antiviral potential of AgNPs is further enhanced by their bactericidal effect against important multi-drug-resistant bacteria, such as methicillin-resistant *Streptococcus aureus* (MRSA), *Pseudomonas aeruginosa*, ampicillin-resistant *Escherichia coli* O157:H7, and erythromycin-resistant *Streptococcus pyogenes* [41]. 

Blosi et al. [50] developed electrospun nanowebs and found a filtration efficiency of 97.7% for nanosized aerosol particles. They found excellent antimicrobial activity as a result of the presence of AgNPs in the electrospun nanowebs. Selvam et al. [51] developed PA/AgNP-based electrospun nanowebs and found a 99% bacterial filtration efficiency, plus they found these nanowebs to be effective against Gram-positive and Gram-negative bacteria. Ahmed et al. [40] suggested that AgNPs can inhibit glycine and alanine of S-proteins along with other proteins from SARS-CoV-2. When incorporated into PPEs, these can effectively protect the attack by the user from SARS-CoV-2. HeiQ^®®^ Viroblock [52] is a silver-based antimicrobial finish that is claimed to be 99.9% effective in 30 min against SARS-CoV-2.

Copper, its alloys, and specific insoluble copper compounds are promising materials for use in fighting against SARS-CoV-2 due to their excellent antiviral and antimicrobial properties [53,54]. Copper(I) iodide has a white color and exhibits high antiviral properties, which is why this material was integrated into masks, filters, and other surfaces [55]. A molecular docking study was also carried out to understand the interaction of Cu(I) with the SARS-CoV-2 main protease [56]. Furthermore, copper nanoparticles, as well as Cu(I) and Cu(II) oxides, were shown to be effective antiviral agents, again after deposition on non-woven polypropylene fabrics that were used as the outer and inner fabric layers in commercial respirators [57]. Kumar et al. [58] developed a reusable and self-sterilizable surgical face mask by spraying a hybrid of shellac (a natural hydrophobic polymer)/copper nanoparticles on a non-woven surgical facemask. They found that it also increased the hydrophobicity of the layer of the face mask, thus stopping the flow of droplets that contain SARS-CoV-2. Jung et al. [59] also developed an antiviral KN94 mask by depositing a 20 nm copper layer using a vacuum deposition method. They found that copper-deposited polypropylene KN94 facemasks have a particle filtration efficiency of 95.1% for NaCl and 91.6% efficiency for paraffin oil. They also showed a 75% reduction in SARS-CoV-2 when the virus came in contact with the copper-coated face mask. Kumar et al. [60] developed a medical face mask by incorporating copper@ZIF-8 core–shell nanowires on a non-woven polypropylene surface. They found a 55% inhibition of virus replication after 48 h by 1 μg of Cu@ZIF-8 NW per well. Despite the fact that copper is an essential trace element for regular metabolism in the human body [61], and the declaration that small amounts of copper-derived particles used for textile coating will not cause health problems during wearing, the potentially toxic and sub-toxic effects of copper nanoparticles, possible allergic reactions, and occurrence of other long-term issues cannot be overlooked [62,63].

Zinc oxide nanoparticles (ZnO NPs) exhibit antibacterial properties against an extensive range of microorganisms, including *Escherichia coli*, *Klebsiella pneumonia*, *Pseudomonas aeruginosa, Pseudomonas vulgaris*, and *Campylobacter jejuni.* In addition, nanoparticles of zinc oxide are considered safe for human contact [41]. Unfortunately, there are not so many studies on the antiviral properties of ZnO NPs. In a recent study, the efficacy of ZnO nanoparticles against SARS-CoV-2 was evaluated. ZnO NPs were produced using an ecofriendly and scalable electrochemical procedure and were fully characterized. Their antiviral activity was tested in vitro against SARS-CoV-2, showing a decrease in viral load between 70% and 90%, as a function of the material’s composition ZnO NPs would be best suited as coatings for commonly touched surfaces [64]. Researchers developed eco-friendly nanowebs of PVA/*Aloe vera*/ZnO that can be used in the inner protective layer of face masks. These composite nanomembranes have excellent antimicrobial properties, killing almost 100% of Gram-positive bacteria and almost 99.2% of Gram-negative bacteria with 4% ZnO nanoparticles in the nanowebs [65]. A similar approach was also taken to obtain an antimicrobial effect with *Aloe vera*/PVA nanofibers [66]. Multifunctional electrospun poly(methyl methacrylate) (PMMA) nanofibers decorated with ZnO nanorods and Ag nanoparticles (PMMA/ZnO Ag NF) were also developed, and these nanowebs showed antimicrobial properties against Gram-positive and Gram-negative bacteria, antiviral properties against SARS-CoV-2, and self-cleaning properties against pollutants [67]. ZnO-loaded PVDF nanofibers have excellent antiviral properties, and 5% ZnO/PVDF nanofibers inhibited virus growth and inhibited viral entry due to the nanoweb structure. These nanowebs have potential applications in filter-embedded half-face respirators [68]. As it is difficult to separate the very thin layer of nanofibers from the electrospinning machine, researchers used a non-woven sheet on the collector so that the nanofibers are collected directly on the surface of the non-woven sheet. Pardo-Figuerez et al. [69] developed and upscaled polyacrylonitrile (PAN)/ZnO nanofibers onto non-woven spunbonded polypropylene (SPP) with different weight bases of 0.4 g/m^2^ and 0.8 g/m^2^. They suggested that a symmetric structure based on SPP/PAN/PAN/SPP was the optimal one, as it reduced SPP consumption while maintaining an FFP2-type filtration efficiency and reducing breathing resistance, especially at high air flow rates, such as those mimicking FFP2 exhalation conditions. They also found excellent antimicrobial properties because of the presence of ZnO nanoparticles in their sheets. 

Titanium dioxide has the required properties, including low toxicity to humans and excellent UV-activated viral inhibition [70]. The antiviral mechanism involves the photocatalytic production of reactive oxygen species (ROS), which are highly unstable and rapidly react with biomolecules in reactions that exchange electrons. This process results in alterations in the structure of biopolymers and lipids, making ROS cytotoxic to a wide variety of organisms [41]. TiO_2_ and polyvinylpyrrolidone nanofibers were also shown to have excellent filtration efficiency. The antimicrobial behavior of TiO_2_ made this nanoweb an excellent material for filtration media, especially in face masks and respirators. PAN/TiO_2_/Ag nanofibers were also developed with a filtration efficiency of particulate matter 2.5. These nanofibers have excellent antimicrobial and UV-resistant properties, making them suitable for filtration media in facemasks and respirators [71].

Iron oxide nanoparticles (IONPs) found interesting medical applications (cancer treatment, magnetic drug targeting, contrast agents for MRI) due to their high biocompatibility and magnetic properties; these nanoparticles were approved by the FDA and various European Union agencies as nanoparticle-based medicines [72]. IONPs exhibit antiviral activity against various viruses due to a broad range of antiviral mechanisms, including ROS generation, lipid peroxidation, and binding to viral surface proteins to impair attachment to host cells. Coating of IONPs with appropriate polymers, including polyvinylpyrrolidone or polyethylene glycol, can enhance their antiviral activity, stability, and safety [41]. A recent theoretical molecular docking study showed the specific interactions of IONPs with the viral glycoproteins of SARS-CoV-2; both maghemite and magnetite interacted efficiently [73]. 

The advantages of metal-based nanoparticles can be summarized as follows: these nanoparticles exhibit a versatile range of antiviral mechanisms that target important viral components, including structural proteins and the lipid envelope. It is expected that the probability of the development of resistance against metal-based NP is low. Different NPs can attack viruses through diverse mechanisms, including oxidative stress, protein disruption, or lipid envelope and capsid damage. In addition, the presence of the antibacterial activity of metal nanoparticles is of high importance. The properties of metal nanoparticles can be finely tuned (e.g., the size and surface modification), even on a larger scale, using green synthesis. The NPs are usually stable [41]; however, their safety in human beings after inhalation is still contradictorily discussed.

Carbon-based nanomaterials, such as graphene, showed utility in numerous applications besides metallic and oxide-based nanoparticles. Graphene and graphene oxide are capable of inhibiting the infective capacity of SARS-CoV-2. When they are incorporated into polyurethane or cotton, the resulting fabric maintains these properties, completely reducing the infectivity of the virus [74]. Complex mechanisms respond to graphene’s capacity for inactive viruses that are dependent on intrinsic material properties and environmental conditions. The physical interaction of graphene or graphene oxide is able to promote structural changes in the virus capsid and envelope [75].

Other forms of carbon, such as carbon nanotubes or carbon dots, also showed their activity. Carbon nanotubes were incorporated into a polyester substrate, giving rise to a filter with efficiency similar to HEPA that can be disinfected by applying resistive heating. In addition to offering other technical opportunities, carbon dots in poly(vinylidene fluoride) membranes confer solar-induced self-sterilization via sunlight absorption and concomitant heat dissipation [76]. However, as for NPs, the safety aspects, e.g., of graphene-related materials is also a matter of current research [77].

The nanomaterials contained in the fabrics possess the risk of leaching during their use and the subsequent management as waste. Studies suggest that the exposure to silver and graphene nanoparticles derived from the continued use of a functionalized mask is much lower than the maximum recommended exposure levels. Estevan et al. [78] indicated systemic (internal) exposure derived from silver nanoparticle facemasks would be between 7.0 × 10^–5^ and 2.8 × 10^–4^ mg/kg bw/day. In addition, conservative systemic no-effect levels between 0.075 and 0.01 mg/kg bw/day were estimated. In the case of graphene, research conducted in the context of the Graphene Flagship and by other investigators in the past several years showed that the hazard potential for different members of the graphene-based materials family may vary considerably and a systematic collection of data on the safety or biocompatibility is yet to be done [78]. However, in the case of its management as waste, this is an aspect to take into account since some of the nanomaterials can show cellular toxicity and damage to the environment [79]. 

To avoid leaching, the nature of the fabric, the surface functionalization of the nanoparticles, and the method of impregnation are important. Among them, the growth of the particles inside the fibers and electrospinning are techniques with the capacity to prevent leaching as the particles are occluded in the polymeric matrix. Gonzalez et al. [80] presented masks prepared from fabrics that included zinc oxide nanoparticles. This work presented a new approach to the synthesis of nanocomposites. It is based on a process known as crescoating (coating via the growth of [-cresco]), which relies on an in situ growth process from the thermal treatment of a dissolved ionic precursor solution. The solution is impregnated into a support material followed by heating so as to begin the nucleation and growth of nanoparticles within the support. The polymers tested were polypropylene and cotton, and relatively heterogeneous distributions of ZnO particles between 5 and 500 nm in diameter were obtained. The authors claimed that the textile-doped nanoparticles thus obtained are non-irritating and hypoallergenic and are capable of achieving a reduction of 3 logs (greater than 99.9%) in the presence of coronavirus, and they maintain their activity even after 100 wash cycles.

Nanomaterials (metallic, carbon-based) can have antiviral properties per se, blocking viral replication and diffusion, or their antiviral properties can be tailored by playing with surface chemistry. However, a huge effort should be undertaken to translate the research into clinics. Several challenges still need to be overcome before their safe use [81]. At the current stage, some nanomaterials were approved only for surface disinfection. For instance, CuNPs have been used in filters for the preparation of highly efficient broad-spectrum antiviral masks [82].

#### 2.2.2. Development of New Methods to Assess the Efficiency of Antiviral Treatments

A limiting factor in the development of novel antiviral technologies is the characterization of their efficacy at inactivating viruses. It is a slow, expensive, but necessary step in the journey of new technology from the laboratory to commercialization. As of July 2022, several established methods exist to characterize the antiviral properties of materials. For instance, the International Standards Organization (ISO) norms 18,184 and 21,702 for the characterization of antiviral properties of porous and non-porous materials prescribe the use of the traditional plaque assay or the TCID50 methods. Another common antiviral characterization method is the immunofluorescence assay, which relies on fluorescently labeled antibodies [83,84,85,86]. A detailed explanation of the established procedures is not within the scope of the present review. However, these require the use of cell cultures; live viruses; trained personnel; incubation times ranging from 2 h to 48 h; and, depending on the targeted virus, laboratory biosafety levels 1, 2, or 3. 

Since the beginning of the pandemic, several studies investigated novel antiviral materials, such as nanoparticles, fucoidans, other naturally extracted compounds, facemasks [15], liquids, and coatings. Most authors characterized their solutions using the plaque assay and/or TCID50 methods [87,88,89,90,91,92]. Less common are cell viability assays (such as MTT and MTS) [93], which estimate the viral load reduction indirectly by observing the cell viability, and polymerase chain reaction (PCR) [84,90], which accurately measures the viral load but presents sampling issues, which makes it impractical in a variety of scenarios. The multitude of studies on antiviral materials and the mentioned drawbacks of existing antiviral characterization procedures highlight the need for novel, fast, and inexpensive antiviral characterization methods.

A few months before the COVID-19 pandemic was declared, Wu Liu et al. reported a microfluidic device that allows for single-cell analysis of viral inhibitors and characterized their mechanisms of action [94]. This is a significant improvement in terms of scientific research possibilities compared with traditional plaque assay experiments, which measure a reduction in infectivity but do not offer further understanding regarding the antiviral mechanisms in play. Nevertheless, this method involves live viruses and cell cultures; therefore, the challenges of availability and ease of use remain unaddressed. 

Angel Serrano-Aroca suggested using the bacteriophage phi 6 as a surrogate for enveloped viruses (such as SARS-CoV-2) [95]. This would allow researchers without access to biosafety level 3 facilities to perform antiviral characterization and significantly facilitate antiviral developments. However, it does not represent a change in methodology and would still involve the expensive and time-consuming procedure of a plaque assay. 

Finally, Furer et al. recently proposed an inactivated virus system (InViS), which can detect viral envelope disintegration via simple fluorescent measurement [36]. The method is based on an inactivated influenza virus labeled with a fluorescent dye, which is released upon the disintegration of the viral envelope. It does not require specific safety protocols, as the virus is noninfectious; the measurement is a fast (less than 15 min) and inexpensive way to characterize antiviral properties, making it ideal for the rapid testing of all layers of a mask separately (Figure 1D). The InViS system is sensible to viral disintegration, which is one of several ways a virus may be inactivated, and can, therefore, only be used to characterize families of materials that take advantage of this specific mechanism. This method’s advantages could complement the established ISO procedures to facilitate the development of novel antiviral technologies by greatly reducing the feedback loop between laboratory solutions and their antiviral characterization.

The development of new methods to assess the efficiency of antiviral treatments should bring us one step closer to determining the filtration properties of medical and technical filtration systems against real viruses in a more relevant exposure scenario. It will facilitate the fast, cheap, and safe pre-screening of a large number of materials and surfaces for potential antiviral properties and will be a valuable tool to support the development of novel antiviral materials, coatings, and facemasks.

### 2.3. PPE Disposal—An Environmental Issue

The COVID-19 pandemic has affected not only human health but also the environment due to the large volume of discarded personal protective equipment (PPE). While personal protective equipment, especially face masks, was recognized as an effective measure to prevent the spread of the virus that causes COVID-19, the remarkable increase in the global usage of face masks and their inappropriate disposal has led to a serious environmental challenge [96,97,98,99,100,101]. Many countries and the World Health Organization (WHO) issued regulations and guidelines on the waste management of PPE and plastics [102] and accordingly handle the PPE discarded by hospitals; however, the main problem is due to the wide use of face masks by ordinary citizens and their careless disposal. 

The improper disposal of face masks not only spreads the disease but also negatively affects the environment. Various environmental risks that originate from the face masks’ inappropriate disposal were recognized and discussed in recent papers, which also proposed possible mitigation strategies. Carelessly discarded face masks can be fatal to the ecosystem; the dyes, inks, and additives leaching from the PPE pose a risk to human health and the environment; and their degradation contributes to microplastic pollution. Face masks can also become a substrate for microorganisms. Additionally, face masks themselves can also act as pollutant carriers and provide a stable environment for pathogenic bacteria and viruses to propagate further since they can adsorb heavy metals and organics [96,97,98,101,103]. All these risks require urgent zero-waste approaches, which may generally be based upon reuse, appropriate disposal, recycling, incineration, and design for recycling, as presented in Figure 2.

Reusable face masks should be promoted instead of single-use face masks in order to effectively extend the service time of masks. Some authors investigated simple reuse approaches [104,105]. Wang et al. investigated a method consisting of combined hot water decontamination with charge regeneration to recover the filtration effects of face masks. The face masks were soaked in hot water at a temperature greater than 56 °C; for 30 min and then dried using an ordinary household hair dryer to recharge the masks with an electrostatic charge to recover their filtration function. The tests conducted on three kinds of typical masks (disposable medical masks, surgical masks, and KN95-grade masks) showed that the filtration efficiencies of the regenerated masks were almost maintained and met the requirements of the respective standards [104]. Zhong et al. developed a dual-mode laser-induced forward transfer method for depositing few-layer graphene onto low-melting temperature non-woven masks. The coated masks were superhydrophobic and the surface temperature of the functional mask quickly increased to over 80 °C, making the masks reusable after sunlight sterilization [105].

At the end of the service life of the face masks, it is important to control the disposal of face masks according to the regulations to ensure that their release into the environment is prevented and directed to the appropriate valorization routes [102,106]. The main component of the face masks is polypropylene [102,107,108], and the recycling routes already applied for the valorization of the polypropylene [109] are also applicable for the valorization of the face masks. The WHO recommends that used face masks should be pre-treated with disinfectants and then incinerated [109] and it has become the most widely used technology in many countries [102]. Incineration converts waste into heat. Plastic waste can be considered a cheap source of energy, as its recoverable energy contents are very high (46 J g^−1^ for PP) compared with other materials [110]. When the collection, sorting, and separation of plastic waste are difficult or not economically viable, or the waste is toxic or hazardous to handle and cannot be sustainably recycled, the best option stands as incineration to recover the chemical energy stored in plastic waste as in the form of thermal energy. However, it is considered ecologically unacceptable due to undesired emissions in the form of toxic and noxious dioxins that should be carefully monitored and requires advanced pollution control measures; these emissions also led to strong societal opposition [111,112,113]. Converting PPE waste into liquid fuel via pyrolysis is a promising alternative waste management approach [113,114,115,116]. Pyrolysis is a thermal process that breaks down the plastic waste into a mixture of products similar to the ones obtained by fractional distillation of crude oil, which range from refinery gases through gasoline/naphtha and diesel to immobile residues in the absence of oxygen (usually in an inert atmosphere) [117]. During pyrolysis, plastic materials are treated at elevated temperatures between 300 °C and 700 °C, which can facilitate the disinfection of contaminated PPE and face masks. Aragaw and Mekonnen demonstrated that used face masks and gloves could be transformed to fuel via pyrolysis at 400 °C for 1 h [113]. Value-added chemicals, such as aromatic compounds, were obtained from face masks in high amounts through catalytic pyrolysis using different types of zeolites as the catalysts [114]. Recently, researchers produced syngas and C_1-2_ hydrocarbon from face masks by using CO_2_-assisted pyrolysis [115]. Mondal suggested converting the PPE waste into pyrolyzed oil and gas for various automobile applications and char in cement industries using pyrolysis by employing solar thermal energy as the heating source [116]. Carbonization, which is a similar process to pyrolysis, is widely applied for converting polymer waste to different valuable carbon materials, such as activated carbon, carbon fiber, carbon sphere, graphite, and carbon nanomaterials, which can be utilized in various applications [118,119]; furthermore, the high temperatures involved in the carbonization process can act as a disinfectant and destroy viruses and other contaminants. Hu and Lin successfully prepared a carbon electrode material with a dense hollow fiber porous structure by combining sulfonation and carbonization for supercapacitor applications [120]. Yuwen et al. obtained porous carbon materials from discarded COVID-19 masks via microwave solvothermal carbonization for use in lithium–sulfur batteries [119]. CNTs were produced from waste face masks via catalytic pyrolysis using Ni–Fe bimetallic catalysts for use as supercapacitor electrode materials [121]. Gasification is another alternative for the valorization of mixed and/or contaminated polymeric waste, where the plastic waste is reacted with a gasifying agent (e.g., steam, oxygen, and air) at a high temperature of approximately 500–1300 °C to produce H_2_, synthesis gas, or syngas [122,123]. Catalytic gasification over Ni-loaded ZSM-5 type zeolites by using steam as the gasifying agent was suggested by Farooq et al. to valorize COVID-19 face masks. Ni-loaded zeolite catalysts not only suppressed the formation of hazardous substances but also enhanced the production of hydrogen from the face masks [124]. As physical methods, extrusion and solvent-based recycling can potentially increase the recycling of discarded face masks. Battegazzore et al. directly fed shredded face masks into an extruder to produce low-cost thin films [125]. A solvent-based recycling technique, as a promising alternative, allows for the reduction of the volume of the plastics; removal of additives or impurities; possibility of precipitating the polymer in different forms for further processing; recovery of high-quality polymers, which can be used for any kind of application; and preservation of the value-added during polymerization, as the polymer itself is obtained rather than monomers or some chemicals. Moreover, it has the potential to deal with mixtures of polymers based on the selective dissolution ability of the solvent used [126,127,128]. The ISOPREP project is currently piloting an innovative solvent-based technology for recycling end-of-life polypropylene products back into virgin-quality polypropylene using a green solvent [129]. On the other hand, using solvents under supercritical conditions presents many advantages in the chemical recycling, solvent-based recycling, and purification of polymers [130,131]. In the United States, P&G partnered with PureCycle Technologies to pilot a method for purifying PP from mixed post-consumer and post-industrial PP waste under supercritical conditions. Their patented process is a purification method that can remove the color, odor, and impurities from waste PP [107]. Considering the recent studies showing that SARS-CoV-2 activity is significantly reduced at 75 ℃ after 15 min [109,132], the solvent-based recycling processes taking place at higher temperatures are expected to facilitate the disinfection of PPE. 

Some researchers investigated the repurposing of discarded face masks in different applications. Zhong et al. suggested using the masks for solar-driven desalination with outstanding salt-rejection performance for long-term use [105]. The incorporation of fibers obtained from plastic waste into concrete, asphalt, and bricks was also investigated [133,134,135]. The waste face masks in fiber or crushed form produced environmentally friendly and affordable green concrete. One percent addition of waste mask fibers was found to increase the compressive and tensile strength, reduce chloride permeability, and increase freeze–thaw resistance of concrete [133]. It was also shown that the incorporation of shredded face masks into hot melt asphalt increased the stiffness and improved the adhesion between the aggregates, resulting in higher pavement resistance to the traffic load [134]. Saberian et al. used face mask waste to prepare pavement base/sub-base with improved ductility, flexibility, and compressive strength [135]. 

Although existing recycling methods mitigate the concerns regarding the waste management of discarded PPE and face masks to some degree, producing and using PPE made up of a single type of material will contribute to the recyclability of the products. Additionally, innovations in materials and fabrication should be encouraged to facilitate biodegradable products, which fully degrade without releasing any harmful by-products.

## 3. Smart Textiles for Monitoring and Sensing as Part of the Telemedicine 

Telemedicine (TM) enables ease of communicating health-related information with low cost and fast accessibility by using the Internet and related technologies [136]. Telemedicine, the importance and need of which is understood in a pandemic period, such as that presented by COVID-19, is a technology that can help both patients and healthcare professionals by minimizing contact with other acute patients and patients with mild symptoms to obtain the supportive care they need. This technology, which is used to maintain and monitor patient health, can benefit from different methods such as online consultations, remote monitoring/scanning, sensors, and chatbots [137]. Since the 1990s, the use of smart textiles, which emerged with the use of auxiliary technologies and chemicals together with textiles, has increased in the health sector, as well as in other sectors [138,139,140,141,142,143,144,145]. Smart textiles can play an important role in telemedicine applications with application areas such as tactile sensing, imaging, communication, energy transmission, and regulating body temperature and humidity [146,147].

As well as providing medical and economic benefits ranging from disease prevention, improved clinical outcomes, and quality of life to increased productivity, reduced healthcare burden, and reduced healthcare costs, such textiles can provide insight into a person’s physiological state and can be used for on-site clinical monitoring and intervention [140]. Smart textiles offer possible solutions for the development and use of personalized healthcare. 

Another issue is signal processing and monitoring of the obtained signals. Because of consumer processing and monitoring requirements, electronic textiles are an intersection set of computer, textile, and electronic sciences (Figure 3) [148].

However, in terms of diagnosing diseases and monitoring patients, although many companies, such as Nextiles (https://www.nextiles.tech, accessed on 1 April 2023), Xenoma (https://xenoma.com/, accessed on 1 April 2023), Skiin (https://skiin.com/ accessed on 1 April 2023), and VTAM (http://www.medes.fr/VTAMN.html, accessed on 1 April 2023) presented alternative suggestions regarding various clothing forms, such as t-shirt vests, the commercialization of smart textile solutions for personalized health services is not in demand [143]. On the other hand, with the COVID-19 era, smart textile manufacturers benefited from washable, flexible textile-based sensors that will not disturb the user. Carpi and de Rossi [139] investigated using electroactive polymer (EAP)-based sensors, actuators, electronic components, and power sources that are implemented as wearable devices for e-textiles. Gao et al. [149] proposed a wearable and smart health monitoring system for wearable telemedicine technology using a fiber mechanical sensor that can monitor health-related physiological signals. By collecting and evaluating the health data to be obtained from the cooperation of smart textile telemedicine, they can help with extracting useful information for future studies [150]. Qiu et al. [151] reported a durable and nondisposable transparent graphene skin electrode for detecting electrophysiological signals, which was fabricated via semi-embedding highly graphitized electrospun fiber/monolayer graphene (GFG) into a soft elastomer, and can be used to reliably collect vital biometric signals, such as electrocardiogram (ECG), surface electromyogram (sEMG), and electroencephalogram (EEG) signals. Kim et al. [152] reported a conformable sensory interface that can be attached to the inside of any user-supplied face mask and used to monitor signals related to infectious diseases, environmental conditions, and wear status of the face mask. Multimodal signals from the sensory face mask are wirelessly transmitted to a server through a custom-made mobile app. The system can simultaneously monitor multiple signals, including skin temperature, humidity, verbal activity, breathing pattern, and fit status of the face mask. The authors also developed a machine learning algorithm that can be used to reliably decode the face mask position.

Electronic textiles (e-textiles), which are formed by the adaptation of conventional electronics to fabric with mechanical applications, are widely used in the medical and health sector [153,154,155]. However, the size, weight, and inflexible structure of the integrated electronics are the main problems in adapting it to the body ergonomically. In order to eliminate this problem, soft, flexible, printed electronics that will provide comfort for the user are preferred for use in e-textile applications [156]. Printed electronics are systems in which electronic devices are manufactured using new materials, such as functional inks, to print on various substrates [157]. Printed electronics are ideal for patient monitoring in healthcare applications, such as wound monitoring, glucose monitoring, continuous ECG, and temperature measurement. Printed electrodes on textiles are suspected to also be used for neuromuscular stimulation (NMES) to aid in muscle strengthening and rehabilitation [158]. Clothing with printed pressure sensors is recommended to monitor the movements of patients and vulnerable elderly living alone in social care facilities or at home [158]. In addition, it is claimed that it can contribute to preventive health services, as well as patient monitoring functions.

Another key challenge in textile sensors is to adequately solve the hysteresis for more broad and exacting applications. Chen et al. [159] tried to solve this hysteretic issue through the structural design of yarns to provide a twisting force. The authors successfully obtained ultrafast response/recovery flexible piezoresistive sensors with DNA-like double helix yarns for epidermal pulse monitoring with an outstanding recovery index and relaxation time.

### 3.1. Temperature Monitoring—Fiber-Based Sensors

Temperature measurement is very important for COVID-19 detection and has been widely used as a first indication of SARS-CoV-2 infection. Body temperature measurement has become an essential part of keeping the public safe, usually by using thermometers and thermal imaging. The use of textile-based temperature sensors that would continuously measure the body temperature of patients would significantly facilitate the work of health workers. Many researchers already offered fiber-based sensors for continuous body temperature monitoring that can be intended for COVID-19 patients. Rajan et al. [160] fabricated and characterized graphene-based textiles for temperature sensing. They demonstrated that despite the scalability of graphene inks, graphene grown via chemical vapor deposition is more suitable for temperature sensors and concluded that these sensors have a potential application in continuous measurement of the human body temperature while being integrated into garments or of the ambient temperature while being integrated into upholstery. Chen et al. [161] obtained temperature-sensitive fluorescent nanoparticle-doped stretchable fluorescent optical fiber that exhibits stable temperature-sensing in the range of −10 to 60 °C with an uncertainty as low as ±0.23 °C and a relative sensitivity of 1.3% °C^–1^, even when it is subjected to large strain up to 40%. They demonstrated sensor-integrated wearable masks and gloves, which can simultaneously measure physiological thermal changes and the movement of the wrist joint.

### 3.2. Electrospun Conducting Polymeric Composite Nanofibers for Sensing

The best-known and most studied conductive polymers are polyaniline (PANI), polypyrrole (PPy) and their derivatives, and poly(3,4-ethylenedioxythiophene) (PEDOT) as one of the polythiophene (PTh) derivatives due to their wide range of applications. Commercially available smart textile products where polymers have a crucial role in their development are medical textiles, protective clothing, touchscreen displays, flexible fabric keyboards, and sensors for various applications. 

Electrospun nanofibers are good candidates for sensitive gas sensors due to the improved surface-area-to-volume ratios of coatings [162]. The high surface area led to higher sensitivity and fast response time. Polyaniline (PANI) nanofiber humidity sensors were produced by electrospinning from the N,N-dimethylformamide solution of PANI, poly(vinyl butyral) (PVB), and PEO [163]. PANI nanofibers with some beads and a small content of PEO revealed high sensitivity, fast response, and small hysteresis because beads could help to improve adhesion to the electrode (which enhances electrical contact and sensing ability), and PEO helped to increase the hydrophilicity of the PANI nanofibers and humidity responses. PANI–polyvinyl pyrrolidone (PVP) composite fibers were prepared for NO_2_ sensing and these mats were reported as a good candidate for this application [164]. PANI–nylon-6 blend nanofiber mats were prepared to determine organic compounds with the advantages of suitable sensitivity and reproducibility [21]. PANI-coated PMMA nanofibers were also used for gas sensing [165]. 

Electrospun fibers are potential candidates for applications in chemosensor fibrous materials [166], nanocomposite materials for producing scaffolds for tissue engineering [167,168,169], drug delivery systems [170], enzyme immobilization supporting active coatings [171], and sensors and actuators [172,173]. The presence of functional and reactive carboxylic acid groups in poly(m-anthranilic acid) (P3ANA) compared with PANA makes it a valuable material with its reactive end groups. Nanofibers ofP3ANA blends with polyacrylonitrile (PAN) obtained using an electrospinning technique could be useful for post-polymerization functionalization on the surface [174]. For the surgical face masks, the main protection mechanism entailed in particulate elimination via fibrous spectrum includes electrostatic attraction, inertial affectation, gravitational sedimentation, and diffusion.

SARS-CoV-2 is classified under the beta-COVs category, spherical, and with a size within the range of 65–125 nm, thereby affirming the importance of developing a highly effective filtration mask. Generally, air filters are aligned into two main groups, namely, membranes and depth filters. Depth filters are fabricated from glass wool, cellulose, and glass fibers, while the filtering mechanism is focused on sedimentation, diffusion, impaction, interception, or electrostatic attraction. Depth filters attain air stabilization by retaining the particulates within them instead of on their surface. Contrastingly, membrane-oriented filters are composed of a thin, porously oriented polymeric membrane in which the filtration mechanism is focused on straining. Hence, the pore size is lesser in comparison with the particle sizes, resulting in effective filtration. Nevertheless, a major challenge arising regarding these filters is the formation of cake due to filtrate agglomeration on the membrane surface, thereby blocking and hindering filtrate passage via the membrane. Therefore, an antifouling mechanism is essential to the enhancement of filter device cleaning [175]. Hence, to overcome the aforementioned challenges, a nanoporous membrane composed of 5 nm pores was constructed and attached to a reusable N95 mask with the capability of replacement after each usage. The porously inclined membrane is based on a naturally occurring hydrophobic polymeric matrix. The droplets come close within the mask, rolling and sliding across the mask because of the increased angle of inclination of the membrane when wearing the face mask. Hence, the nanoporous membranes capable of being attached atop an N95 mask give protection by inhibiting SARS-CoV-2.

### 3.3. Nanogenerator-Based Textile Sensors

Nanogenerators, which are energy-generating units that convert mechanical or thermal energy into electrical energy, have also been a trending device class over the last decade [176]. Further, nanogenerators are commonly based on polymeric materials thanks to the flexibility advantage of polymers. In other words, nanogenerators can be easily adapted to textile applications for sensing or energy harvesting. In the energy-transforming process, nanogenerators use three main principles: piezoelectricity (pressure stimulated), triboelectricity (tactile stimulated), and pyroelectricity (thermal stimulated) [177]. Current applications of nanogenerator-based medical sensors are for detecting coughing, heartbeats, cardiovascular status, body temperature and sweat, respiration, and muscle functionality [178,179,180].

The importance of medical detection was clearly seen during the COVID-19 pandemic for various purposes. Infection diagnosis, body temperature, respiration, and oxygen saturation measurements and/or tests were the most important topics during the pandemic [181,182]. However, breath sensing is the most important case due to severe respiratory distress during COVID-19 illness. 

As one of the main medical sensor types for medical applications, respiration monitoring brings two big advantages: respiration health detection and respiration chemical detection [183]. Although chemical analysis of breath is a common method for ammonia, alcohol, acetone, humidity, and CO_2_ detection, it is not a convenient method for COVID-19 diagnosis or control applications yet. Respiration health detection can be defined as the detection of unhealthy breathing during inhale–exhale cycles by analyzing the output signals. Because of this, respiration health monitoring is a physical method, and piezoelectric and triboelectric principles are commonly used for this type of sensor. In particular, piezoelectric systems are quite suitable for respiration sensing thanks to their low stimulation threshold [184]. Additionally, pyroelectric materials also can be used for breath sensors [185,186]. 

Healthcare masks are the biggest candidates for textile-based respiration sensors, as expected. The airflow force of breathing is the driving force for mask-based respiration sensors using the pyroelectric or piezoelectric effect (Figure 4a). This also means that the mechanical magnitude of breath can be measured sensitively with current technology. Triboelectric, pyroelectric, or piezoelectric components, which are also sensing components of a mask system, can be integrated into a commercial mask (Figure 4b) [187]. Similarly, pyroelectric-based respiratory sensors can be easily implanted onto face masks. In the pyroelectric effect, pyroelectric material gets polarized with heating–cooling cycles, and electrical flow occurs. The required heating–cooling effect can be easily provided by the inhale–exhale cycles for a respiration-sensing mask [185]. Besides the popularity of face-mask-integrated sensors, another type of breath sensor was reported [183], where the sensing component is strictly mounted onto the chest of the patient.

Besides all the mentioned sensing mechanisms, obtained data (from the sensor) should be processed. In this regard, personalized breathing patterns of patients in healthy a condition should be obtained. This pattern is used as the reference breathing behavior of the person. Thus, unhealthy breathing cases or rates of unhealthiness can be easily detected [183].

### 3.4. Perspectives and Challenges

The role that smart textiles are going to play in all phases of medical treatment, namely, prevention, immediate care, and rehabilitation, is indisputable. Integrating smart clothing into our lives is the next step. One likely future scenario is that as the field of fibertronics becomes more mature, hybrid structures will incorporate more electronic functionality at the fiber level until we eventually end up with electronic textiles where all advanced electronic functions are embedded in the textile fibers [188]. They will non-invasively monitor a wide range of body parameters, ultimately enabling comprehensive medical diagnostics and performance assessment. However, their acceptance by the medical community will require extensive and successful validation in human trials and an improved understanding of the clinical relevance of the information provided [189].

In addition to the challenges related to smart textiles and their production, design, device operation, relevance, stability, and implementation, which are related to the research development, the complexity of the subject also gives rise to other challenges. One challenge is related to telemedicine and data protection, while the other is related to the disposal of smart textiles, as there is currently no regulation to treat them as textile or electronic waste [190].

## 4. New Products on the Market

The smart textiles market has been witnessing significant growth over the last few years as a result of the progress in big data, artificial intelligence, miniaturization of electronics, and the growing need and interest. According to a market research report, the smart textile market is expected to grow from USD 2.3 billion in 2021 to USD 6.6 billion by 2026; it is expected to grow at a CAGR of 23.2% during the forecast period [191]. According to the report of Mordor Intelligence, the wearable medical devices market, valued at USD 27.91 billion in 2019, is expected to reach USD 74.03 billion by 2025 while exhibiting a CAGR of 17.65% over the forecast period of 2020 to 2025.

The COVID-19 pandemic has increased the popularity of smart textiles, especially in preventive healthcare, with applications ranging from illness diagnosis to wellness monitoring [191,192]. During the pandemic, the restriction on social distancing, home confinement, and remote work reduced the physical activity time in all age groups by at least 30–50% after the lockdown [193]. This situation had unfavorable effects on physical and mental health. Furthermore, faced with the sudden outbreak of the COVID-19 pandemic, the hospitals were saturated and unprepared to offer their services to the huge number of patients presented.

**Figure 5 healthcare-11-01115-f005:**
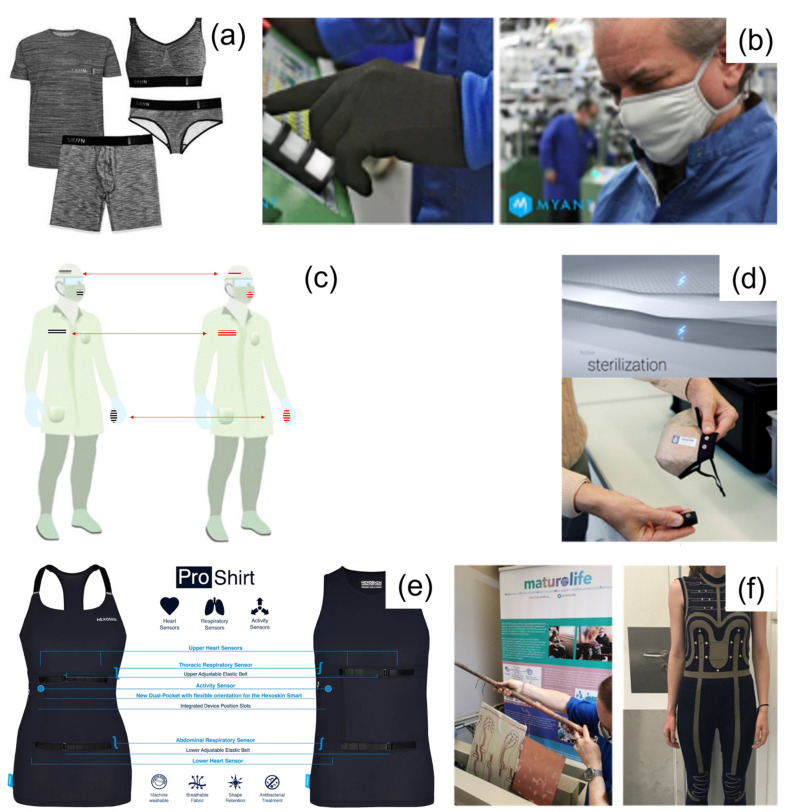
Skiin biosensing garment by Myant (**a**) [194]; Myant gloves and masks (**b**) [195] (reproduced with permission from Myant); sensor-integrated protective wearables for the early detection of SARS-CoV-2 before (left) and after contamination (right) (**c**) [196]; Osmotex sterilizer: structure [197] (top), mask prototype [189] (bottom) (**d**) (reproduced with permission from Osmotex); Hexoskin proshirts (**e**) [198] (reproduced with permission from Hexoskin); and Maturolife project shoe sole prototyping for walking pattern (left) and E-Fashion^2^ project movement sensor prototyping (right) (**f**) [199] (reproduced with permission from IFTH).

Due to the outcomes of the crisis, the interest in continuous vital signs monitoring with smart textiles for the early diagnosis and remote treatment of COVID-19 but also for the remote diagnosis of other illnesses regarding the overcrowding of hospitals has increased. Smart textiles are mostly used to monitor vital data, such as a patient’s heartbeat, ECG, carotid pulse, respiration, breathing pattern, skin temperature, skin impedance, and physical activity. By including the big data, it is possible to engage in early diagnostics or prevention, as well as better understand the effects and personalize the treatments [200].

Some projects propose textile-based solutions for monitoring key physiological data for individuals from a distance and communicating those data to healthcare professionals to detect health changes and support medical decisions. These solutions may apply to individuals with and without COVID-19. 

One of these mentioned projects is the collaboration of Nanowear with Hackensack Meridian Health Systems and the Maimonides Medical Center in the USA [201]. The clinical trial collaboration is focused on the remote diagnostic monitoring of confirmed or suspected COVID-19 patients with Nanowear’s fabric-based nanosensors. This undergarment captures and algorithmically scores phonocardiography, impedance cardiography by measuring stroke volume and cardiac output, multichannel ECG assessing heart rate and heart rate variability, respiratory rate, thoracic impedance, activity, and posture [202].

Another example is from Toronto-based textile technology company Myant. They propose a wearable textile solution (Figure 5a) [194] for remotely monitoring COVID-19 symptoms thanks to their ability to record a patient’s ECG, chest motion and sound with breathing, and skin temperature.

During the pandemic, besides the textile sensors, Myant also developed antibacterial and antiviral knitted facemasks and gloves treated with copper (Figure 5b) [195]. The gloves are suitable for protection and interaction with capacitive touchscreens, thanks to the conductive fibers. These products propose an additional magnitude of protection, as they minimize viral exposure and maximize efficacy.

Austrian national project “V-Protect Plus” [203], owned by the industrial partner Grabher Group (GG) [204] cooperating with V-Trion GmbH [205], effectively responded by developing self-cleaning, droplet-proof protective functional facemasks to defend against SARS-CoV-2 since the very early phase of the current pandemic in 2020 [206]. 

The application of textile and flexible noninvasive biosensors was found to be effective in providing on-body health-related data, such as the content of lactate and pH, in real time [207]. Therefore, integrating biosensors for pathogenic detection, such as SARS-CoV-2, into wearable protective cloth is strategically important for current times. Pathogen detection from the environment is obtained via a biochemical reaction that changes the electrochemical and functional properties. Several techniques improved on-site monitoring healthcare applications, such as electrochemical detection, optical detection, and colorimetric techniques [208,209,210]. Among these techniques, the most employed detection technique is based on colorimetric detection (Figure 5c). This technique can easily analyze the results by a color change produced via ligand analyte interaction. The most common substrate materials used to fabricate flexible biosensors are polymer, paper, and textile. Paper-based biosensors have been shown great interest due to their foldability, portability, and accessibility; furthermore, this kind of biosensor can be easily modified with biomolecules and nanomaterials, which helps to improve the sensitivity. Despite the high potential of paper-based biosensor applications, there are several drawbacks regarding analytical performance, such as its accuracy, sensitivity, and detection of multiple biomarkers simultaneously. The other important flexible biosensors use textiles as substrate materials, which have a promising prospect in the area of wearable sensors due to the knittable nature of the fabric. However, some other innovative techniques, such as a textile-based lateral flow (LF) strip, which combines nanoparticles with conventional chromatographic separation, thus creating a moisture-responsive layer with pathogen-sensing nanomaterials or via a direct coating of a 3D moisture-containing hydrogel network layer. 

A Swiss-based company Osmotex developed a power-controlled self-disinfecting textile called Osmotex sterilizer [197], which is a three-layer electrochemical textile (Figure 5d). This hygroscopic sandwich structure consists of a membrane between two conductive textiles and disinfects itself with an electric potential. This technology is developed in response to the COVID-19 pandemic, but it has many application areas, such as high-touch surfaces in public spaces and sheets and beds in hospitals and hotels.

The Quebec company Hexoskin also develops washable, comfortable smart garments, including textile sensors that are embedded into comfortable garments for precise and continuous cardiac, respiratory, and activity monitoring (Figure 5e). Hexoskin biometric shirts can continuously measure respiration effort in addition to cardiac activity to observe the evolution of the disease and its effects on lung function [198].

European textile research laboratories are also carrying out many projects to monitor body and environmental parameters to respond to market needs. Different textile processes insert smart and interactive functionalities into textiles for various applications. IFTH took part in projects that included design and end-user acceptance, as well as technological innovation. One of these projects was the Maturolife H2020 project [199], which aimed to develop lightweight, breathable, and washable textile sensors with an electroless metallization process to make urban living for older people more independent and fashionable (Figure 5f). The technology used in the E-fashion^2^ project, which involved knitted clothing with flexible, breathable textile sensors, can also be used to monitor body parameters for therapeutic areas, such as respiratory, somnology, and physical performance.

Despite the concerns about regularity and reimbursement issues, there is a growing acceptance of smart clothing, which is paving the way for the market. These garments allow patients to recover comfortably from home, help the aging population, efficiently track chronic conditions to inform preventative healthcare better, and advise better patient care with accurate diagnostic data thanks to AI and deep-learning-enabled diagnostics [211]. 

## 5. Conclusions

The emergence of the COVID-19 pandemic has significantly affected the development of advanced and smart textiles, especially in personal protection and telemedicine. Certainly, accelerated by the pandemic, electrospinning and similar methods were identified as promising techniques to produce nanofibrous filter materials, which not only satisfy the filtration efficiency and permeability criteria but also offer opportunities such as transparency, reuse after washing, and disinfection. The research was also focused on the antiviral functionalization of PPE, mainly by incorporating metallic or carbon-based nanoparticles. This certainly raised the question of the disposal of this type of product at the end of its life cycle, which further emphasized the already existing problems with the disposal of PPE, especially the masks that were intensively used by everyone. Although existing recycling methods mitigate the concerns regarding the waste management of discarded PPE and face masks to some degree, producing and using PPE made up of a single type of material will contribute to the recyclability of the products. Thus, innovations in materials and fabrication should be encouraged to facilitate biodegradable products, which fully degrade without releasing any harmful by-products.

Smart textiles with the role of sensors and actuators, as a part of telemedicine, also experienced a research and development boom after the SARS-CoV-2 outbreak. Despite the concerns about regularity and reimbursement issues, there is a growing acceptance of smart clothing, which is paving the way for market expansion. These garments allow patients to recover comfortably from home, help the aging population, track efficiently chronic conditions to inform preventative healthcare better, and advise better patient care with accurate diagnostic data thanks to AI and deep-learning-enabled diagnostics. These solutions can also reduce costs, as they increase health insights.

However, there are still challenges that need to be overcome: these are related to the production, design, device operation, relevance, stability, implementation, data interpretation, etc., as well as others related to data security and disposal.

Considering all the challenges, the optimal outcome is to emerge from the crisis with a clearer vision of how to develop smart and advanced textiles to reap the benefits while avoiding or minimizing negative effects and potential abuse.

## Figures and Tables

**Figure 1 healthcare-11-01115-f001:**
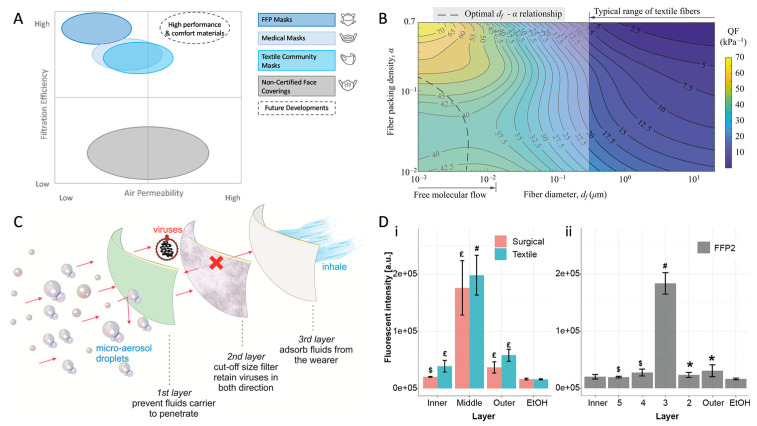
Filtration efficiency and air permeability as driving factors for community mask development. Categorization of face mask types (**A**) [15], air permeability limited by fiber architecture (**B**) [25], the established layer structure of community masks with nanofibrous non-woven fabrics as a filter (**C**) [20], and viral distribution per layer after simulated use in common face mask types (**D**) [36]. (**A**,**B**,**D**) Reproduced under CC BY 4.0 (**C**) Reproduced with permission from the American Physical Society.

**Figure 2 healthcare-11-01115-f002:**
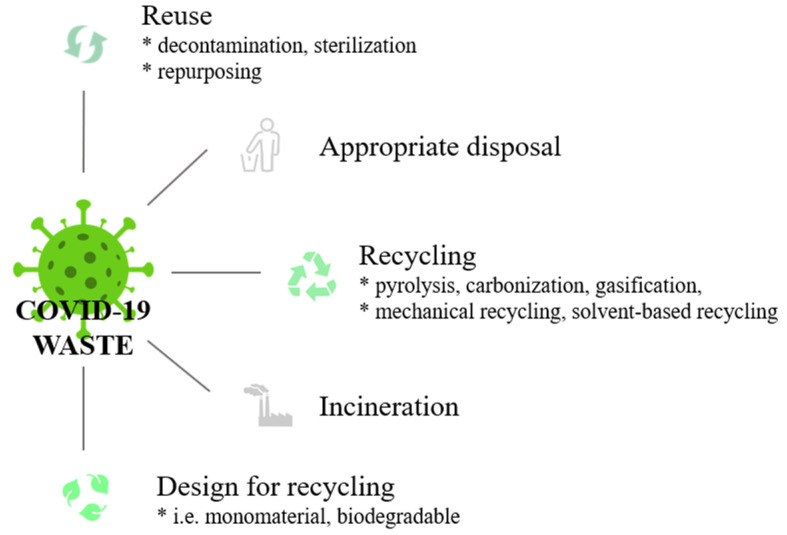
Zero-waste approaches to address COVID-19 waste.

**Figure 3 healthcare-11-01115-f003:**
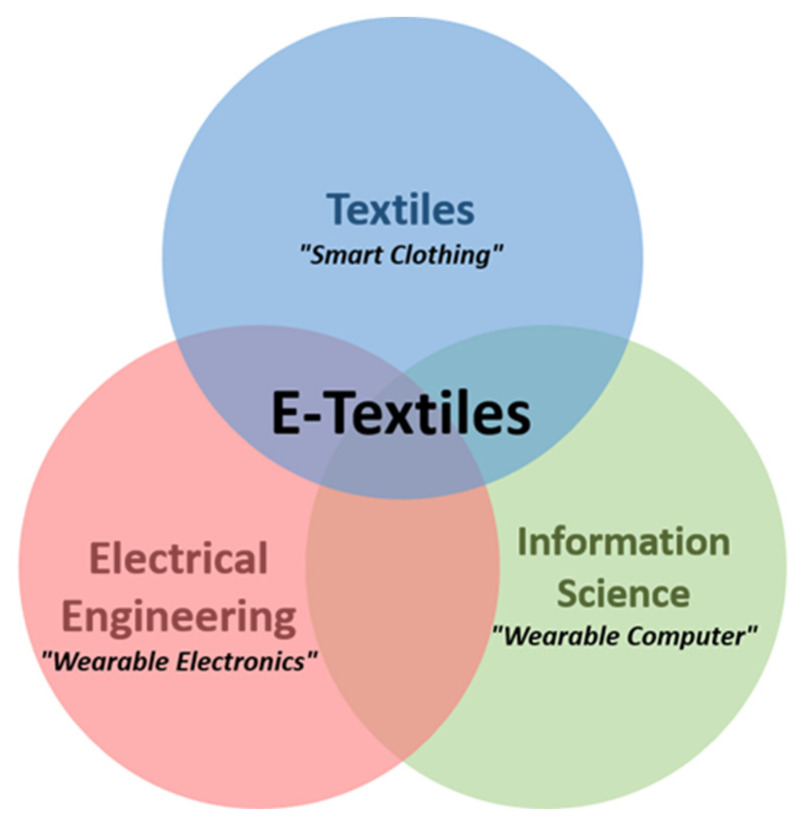
Classification of supersets of electronic textiles [148]. Reproduced with permission from Elsevier.

**Figure 4 healthcare-11-01115-f004:**
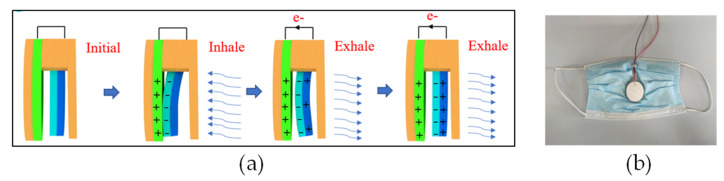
(**a**) Working mechanism of a mask−based respiration sensor [187]. Reproduced with permission from Elsevier. (**b**) Image of a triboelectric component integrated into a face mask [187]. Reproduced with permission from Elsevier.

**Table 1 healthcare-11-01115-t001:** Performance and requirements of medical face masks as per international standard EN 14683:2019.

Test	Type I	Type II	Type IIR
Bacterial filtration efficiency (%)	≥95	≥98	≥98
Differential pressure (Pa/cm^2^)	<40	<40	<60
Microbial cleanliness (cfu/g)	≤30	≤30	≤30
Splash resistance pressure (kPa)	Not required	Not required	≥16.0

**Table 2 healthcare-11-01115-t002:** Some performance requirements of respirators as per international standard EN 149:2001.

Test	FFP1	FFP2	FFP3
NaCl filtration (%)	80%	95%	99%
Paraffin oil filtration	80%	95%	99%
Breathing resistance (mbar)—inhalation 30 L/min	0.6	0.7	1.0
Breathing resistance (mbar)—inhalation 95 L/min	2.1	2.4	3.0
Breathing resistance (mbar)—exhalation 160 L/min	3.0	3.0	3.0
Flammability	<5 s	<5 s	<5 s

## Data Availability

Not applicable.

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
