# Peer review of "Advanced and Smart Textiles during and after the COVID-19 Pandemic: Issues, Challenges, and Innovations"

_healthcare, 2023, doi:10.3390/healthcare11081115_

Round 1

Reviewer 1 Report

The authors systematically reviewed the recent development of smart textiles that emerged as a response to the outbreak of Covid-19, based on three sections: 1. Textile for personal protective equipment; 2. Smart textile for sensing; 3. Advanced commercialized textile products. Overall, the manuscript is well organized, and it is recommended for publication after minor revision. Some comments are as follows:

Major:

1.        Section 3 is divided into electrospun and nanogenerator-based textile sensors. Although it’s not comprehensive, it’s suggested that authors may briefly introduce textiles for thermal management, which is especially practical during the pandemic period.

2.       It’s suggested authors may cite the following papers in section 2 or 3, ‘Smart Electronic Textiles for Wearable Sensing and Display. Biosensors 2022, 12, 222.’, which has an overlap part regarding smart textiles for sensing; ‘A bioinspired, durable, and nondisposable transparent graphene skin electrode for electrophysiological signal detection." ACS Materials Letters 2.8 (2020): 999-1007.’, in which electrospun phenolic resin fiber was used; ‘A conformable sensory face mask for decoding biological and environmental signals. Nat Electron 5, 794–807 (2022).’, which proposed a smart mask.

Minor:

1.     Line 90-91, please revise ‘28 thousand’ and ‘230 thousand’ to all numbers or all words.

2.               Line 173-178, please revise the description order.

3.               Line 411 and 685, what is ‘Error! Reference source not found’ Please confirm the reference source.'

4. There is no corresponding author.

5. In Author Contributions section, it shows 'funding acquisition', but in Funding section, it shows 'no external funding', which is contradictory.

Author Response

Please see the attachment. Name of the file : Answers to Reviewer 1 comments.docx

Reviewer 2 Report

As attached. Title needs to be amended to better fit the thrust and the contents of the paper.

Author Response

Please see the attachment. Name of the file : Answers to Reviewer 2 comments

Reviewer 3 Report

This review is organized well, and recent advances in advanced and smart textiles are summarized in this review along with critical analysis of the issues, challenges and innovations. This paper can be more attractive if some small flaws are addressed. For example, the advanced and smart textiles by applying nano materials or using various nanofiber materials are reviewed, but the advances of modified textile structures and corresponding studies are not mentioned, which is considered important out of 'materials and structures'. So if the authors briefly review or analyze the current advances of advanced textiles used in medicine based on modified structures, such as yarn structure design (doi.org/10.1002/adma.202104313) and fabric design (doi.org/10.1002/adfm.202210351), the paper will be more comprehensive and interesting.

Author Response

Please see the attachment. Name of the file: Answers to Reviewer 3 comments.docx
